# Exploring the pre-instruction gender gap in physics

Eric Burkholder[1]*, Shima Salehi[2]

1 Department of Physics, Auburn University, Auburn, AL, United States of America, 2 Graduate School of Education, Stanford University, Stanford, CA, United States of America

* ewb0026@auburn.edu

**Data Availability Statement:** We note that we are forbidden by our IRB and the Stanford University Student Data Oversight Committee to share any data publicly that includes student demographic information, even if it is anonymized. This is to reduce the possibility of individual students being

## Abstract

There is a substantial body of work in physics education looking at gender disparities in physics. Recent work has linked gender disparities in college physics course performance to disparities in high school physics preparation, but to our knowledge, the origin of the disparity in high school physics preparation is still underexplored. In a select sample, we found that women on average had lower force and motion conceptual evaluation (FMCE) pre-scores (the FMCE is a short conceptual assessment of Newton's laws), and FMCE pre-score entirely mediated the effects of high school preparation and social-psychological factors on exam performance. The gender gap in FMCE pre-scores could not be explained by differences in the number of physics courses taken in high school. Instead, we find that the gender gap in the FMCE is partially explained by female students' higher levels of general test anxiety. We hypothesize that the format of the FMCE, a timed assessment, triggers stereotype threat in female students despite being a low-stakes assessment. Therefore, instructors and researchers should take care in interpreting the results of such concept inventory scores and should re-think the way they assess understanding of physics concepts. Results of this work aligned with previous findings on gender disparity in timed exams call upon investigating gender equitable assessment formats for evaluating physics knowledge to replace timed assessments, either high or low stakes.

## Introduction

Many demographic groups are currently underrepresented in STEM fields [1,2]. To address this under-representation, we need to make sure the paths to choose and excel in STEM fields are accessible to underrepresented demographic groups [3]. STEM introductory courses are an influential part of this path [4]. Previous works have shown that student performance in introductory STEM significantly affects STEM retention [5]. Furthermore, low academic performance in these courses can also have negative psychological consequences for marginalized students (Nardo et al., in press). One of the main introductory STEM courses is calculus-based physics 1. This course is offered in all STEM higher education institutions with similar course content and a pre-requisite for many STEM fields, and hence a gatekeeper for them. The interviews with students who have left STEM fields in the last two decades have shown that

identified through a constellation of their prior physics experience, gender and race, for example. For more information, contact Corrie Potter (cjpotter@stanford.edu).

**Funding:** The authors received no specific funding for this work.

**Competing interests:** The authors have declared that no competing interests exist.

challenges in this course is one of the main reasons for leaving STEM fields despite their initial aspirations [5].

Firstly, previous studies have shown that the common instructional practices in physics 1, like other introductory courses, are particularly disadvantageous for marginalized students. The strongest predictor of students' performance in these courses, including physics 1, is STEM preparation from high school experiences, referred to as *incoming preparation* hereafter. Many marginalized students attend high schools which are under-resourced particularly in STEM fields [6,7], leading to lower STEM preparation. Salehi et al. [3,8] have shown that while marginalized students underperform in physics 1 across three vastly different institutions, these demographic performance gaps can almost entirely be explained by the students' lower incoming preparation. After controlling for variation in incoming preparation of students, there was no difference in performance of marginalized and over-represented students. In their work, they had two measures of incoming preparation: 1) general incoming preparation as measured by Math SAT/ACT–a standardized college entrance exam in the United States, 2) physics-specific incoming preparation as measured by physics concept inventories, such as the Force and Motion Conceptual Evaluation (FMCE). They show that while general incoming preparation can explain most of the performance gaps for non-white and non-Asian under-represented racial minority (URM) and first-generation students, the gender performance gaps are mainly due to lower physics-specific incoming preparation. For women, lower scores on the math SAT/ACT could explain only about one-third of the gap, with physics-specific incoming preparation explaining the remaining gap.

Secondly, previous studies have shown that high-stakes exams disadvantage women in introductory STEM courses, while in non-exam assessments (e.g., homework, quizzes), if anything, women overperform compared to men [9,10]. These studies have shown that women have higher test anxiety regardless of their incoming preparation and test anxiety negatively affects their exam performance. For men, on the other hand, test anxiety was negatively correlated with their incoming preparation and did not affect their exam performance when controlled for incoming preparation. Similar findings have been replicated across different STEM courses [11]. In this study, we expand on the intersection of these two lines of work. We examine what are the root causes of women's lower physics-specific incoming preparation, and whether the measurement format impacts the gender gap in both incoming preparation and physics 1 performance.

## Theoretical framework

Physics education researchers have long thought about the gender disparities in physics, particularly when it comes to representation and course performance. Indeed, representation is an issue that starts before students enter college, with fewer women on average taking advanced physics courses [12]. Sax et al. [13] discuss several potential explanations for these observed gender disparities: familial expectations and beliefs, K-12 experiences, psychological factors, and perceptions of physics as a field. Familial expectations set by parents' influence on their children's course taking, as well as the careers of the parents, may result in fewer women taking advanced science and math coursework in high school–either due to lack of interest or being discouraged from doing so [14]. Lack of K-12 preparation as well as negative experiences in those environments can also dissuade women from pursuing STEM [15]. These experiences also impact things like self-efficacy and interest at the college level [16].

Most studies of gender in physics have focused primarily on the social psychological aspects of interest and persistence [17] despite the clear evidence that these are deeply intertwined with prior academic experiences and achievement. We adopt a version of Social Cognitive

Career Theory (SCCT) in which learning experiences, background characteristics, contexts, and performance are interrelated and dynamic [3,18]. In this model we hypothesize that background characteristics (socioeconomic status, parental education and careers) influence students' learning experiences. For example, students from lower socioeconomic strata may not have access to the same quality of instruction as students from higher strata. These early learning experiences influence students' psychology such as their motivation, self-efficacy, interest in a subject, test-anxiety. There is then a feedback loop between psychology and learning experiences that impact students throughout their academic careers. This can be reinforced or mitigated by contextual influences.

This theory, along with our prior empirical work, inform our research questions. We assume that students' background characteristics (in this case, gender) will influence their prior academic experiences and social psychological profile. For example, women are less likely to take AP physics courses on average nationally. They may also be more likely to suffer from test anxiety and/or stereotype threat due to early academic experiences. These factors will influence the students' level of preparation for college physics. We then test whether this level of preparation is the sole mediator for future physics achievement, or whether these social-psychological factors and personal characteristics also play some direct role. Our research questions are thus:

1. Are there gender disparities in physics1 performance as measured by exam performance? (RQ1)

2. Are there gender disparities in physics 1 incoming preparation as measured by FMCE pre-test? (RQ2)

3. What learning experience, background characteristics, and contextual factors contribute to these gender disparities (i.e., what are the mediators of gender disparities in exam score and FMCE pre-score?)? (RQ3)

Note that there are a number of studies that have already investigated RQ1 and RQ2. We repeat them for the current sample to provide more context to the results of RQ3.

## Methods

### Data and measures

We collected data from a private, highly selective research-intensive university in the western United States during the years 2018 and 2020. This project was approved by the Stanford University IRB, protocol number 48006. In both years, the FMCE and a background survey were administered for attendance credit at the first recitation section. The background survey asked about students' prior physics and math coursework, and in 2020, also probed several social-psychological constructs based on previously validated surveys [3]. Table 1 summarizes these factors and some previous works examining their effects on STEM performance. The same survey was administered for attendance credit in the final recitation section of the term as well. A binary measure of students' gender was obtained from the office of institutional research for both years. We acknowledged that binary measure of gender has limitations, but that is the only format used in institutional data sets. At the beginning of the course (the time of the FMCE), students gave written consent (checked a box) to allow their data to be used for research purposes. We only included data for students for whom we had consent. All participants were over the age of 18. This study was determined exempt from review by the Stanford University IRB, protocol number 48006.

**Table 1. List of social-psychological constructs probed by the survey and relevant references.**

| Social Psychological Factor | Reference |
|---|---|
| Test Anxiety | Measures feelings of anxiety and unease during exams [19] |
| Growth Mindset | Measures tendency to view intelligence as fixed [20] |
| Sense of Belonging | Measures social integration in the science classroom [9] |
| Self-Efficacy | Measures how well students think they are able to do science [3] |
| Science Identity | Measures how important science is to person's self-esteem [21] |
| Motivation to Learn | Measures whether motivation to learn is performance-motivated or intrinsic [3] |
| Interest in Science (Physics) | Measures whether students find subject material interesting [19] |
| Ethnicity Stereotype Threat | Measures whether they think their peers perceive them differently due to their race [21] |

## Analysis

We used a combination of multivariable linear regression and structural equation modeling (SEM) to investigate the relationships between gender, incoming preparation, social psychological factors and course performance as measured by the final exam score. We first used multivariable linear regression as a tool physics education researchers are more familiar with compared to SEM. Multivariable linear regression allows one to simultaneously investigate the effects of multiple variables on a particular outcome. We were interested in the effects of gender, social psychological factors, and incoming preparation on final exam performance. An example regression model looks like:

$$Exam\ Score = \beta_0 + \beta_1 Incoming\ Preparation + \beta_2 Test\ Anxiety + \beta_3 Gender + \epsilon$$

Compared to multivariable regression, SEM is a more complex method and has the advantage of not only estimating the effects of multiple variables on an outcome, but also allowing to examine the structure of relationships between the independent variables. SEM tests how well a structure, as specified by a set of linear models and model covariances, can predict the observed covariance matrix of the entire data set using maximum-likelihood estimation. An example of a simple SEM model would be:

$$Incoming\ Prepartion = \gamma_0 + \gamma_1 Gender + \epsilon_1$$

$$Exam\ Score = \beta_0 + \beta_1 Gender + \beta_2 Incoming\ Preparation + \epsilon_2$$

In the above example models, while the multivariable regression method only allows one to test whether and to what extent gender predicts exam score, SEM methods allows one to also test different paths for the effect of gender on exam score. In the above SEM model, gender affects exam score through two paths: indirectly through incoming preparation and directly. The indirect effect is that gender predicts level of incoming preparation and the level of incoming preparation in turn predicts student exam score. In other words, incoming preparation mediates the effect of gender on exam score. The SEM model represented as two equations above can also be visually represented as shown in Fig 1. The size of mediation effect can be calculated by multiplying the coefficients of the mediation path, in this case $\gamma_1$ and $\beta_2$.

In this work, we used SEM to examine the mediators of gender differences in exam performance. For more details on SEM analysis, the advantages and shortcomings as well as different tools to conduct such analysis, we refer the reader to Ballen and Salehi [22].

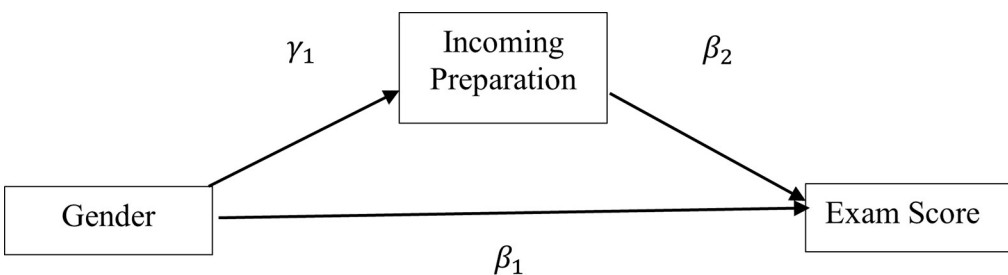

**Fig 1. Visual representation of the structural model explained above.**

## Results

RQ1. Are there gender disparities in physics 1 performance as measured by exam performance?

Table 2 summarizes the results of the regression analysis. First, we examined whether course performance and/or physics incoming preparation measured by FMCE pre-score varied across gender. The first model in Table 2 predicts students' performance only based on gender. As found previously in Salehi et al. [8], Model 1 shows that there is a gender gap of 0.22 standard deviations in exam performance indicating that on average women gained 0.22 standard deviation lower exam score compared to men.

Then we tested how physics 1 performance is impacted by students' gender after controlling for physics incoming preparation, as measured by FMCE pre-score. Based on previous research, we hypothesize that there would be a gender gap in exam grades, but that gap would be explained by differences in FMCE pre-scores. The second model in Table 2 uses both gender and physics prior preparation. The gender gaps captured by Model 1 becomes small and insignificant in Model 2 when controlling for FMCE pre-scores, suggesting that the gap in exam grades can be traced to differences in male and female students' prior physics preparation as measured by FMCE pre-score. This confirms our hypothesis that there exists a gender gap in exam performance, but it can be explained by variation in physics incoming preparation, which aligns with previous studies. We found no interaction between gender and FMCE pre-score as shown in Model 3, suggesting that the relationship between preparation and exam performance is the same for male and female students.

RQ2. Are there gender disparities in physics 1 incoming preparation as measured by FMCE pre-test?

**Table 2. Multivariable linear regression predicting the effects of gender and FMCE pre-scores on exam grades.**

|  | Model 1 | Model 2 | Model 3 |
|---|---|---|---|
| Gender (F = 1) | -0.22* (0.10) | 0.093 (0.093) | 0.095 (0.095) |
| FMCE Pre-score |  | 0.52***(0.046) | 0.47***(0.062) |
| FMCE x Gender |  |  | 0.11 (0.095) |
| R-squared | 0.0092 | 0.27 | 0.25 |
| N | 366 | 366 | 366 |

*** $p < 0.001$

** $p < 0.01$

* $p < 0.05$.

We then examined to what extent FMCE pre-scores depend on gender (see Table 3). We found a gender gap of 0.48 standard deviations on the FMCE, with female students being disadvantaged. This is consistent with previous investigations at this university [8].

RQ3. What learning experiences, backgrounds characteristics, and contextual factors contribute to these gender disparities?

We also examined to what extent FMCE pre-scores depend on gender when controlling for the number and type of physics courses a student took in high school. We hypothesized that the gender gap in FMCE pre-scores is due to gender differences in either the number or the quality of physics courses a student took in high school (Table 3). The gender gap in FMCE pre-score remains unchanged when controlling for high school physics course. This indicates that gender differences in FMCE pre-scores are uncorrelated with the number of physics courses taken in high school for students at this university. This contrasts with reports that female students are less likely to take physics in high school [23]. The size of the gender gap also remains the same when controlling for whether a student had taken an AP mechanics course (AP Physics 1 is an algebra-based college-level physics course, AP Physics C is the calculus-based version of the same course). Overall, these findings, in contrary to some previous speculations, suggest that in this sample the gender variation in incoming preparation as measured by FMCE pre-score is not due to gender differences in the type or the number of high school physics courses. We expect that this is specific to this sample because of the highly-selective nature of the institution.

We next examined gender differences in various social-psychological factors at the beginning of the course (see Fig 2). Female students reported much higher test anxiety than male students as also shown in previous work [9,11] (d = 0.51 ± 0.10, p < 0.001) and a lower sense of belonging (d = 0.31, p < 0.01). However, female students reported higher interest in physics (d = 0.22 ± 0.11, p < 0.05) and a greater sense of physics identity (d = 0.23 ± 0.11, p < 0.05).

Using the SEM model, we tested how these social psychological factors as well as high school physics courses can mediate the effect of gender on FMCE pre-scores (see Fig 3 for the model structure). By all fit indices, this mediation model is a good fit for the data covariance matrix (acceptable ranges: CFI > 0.95, TLI > 0.90, RMSEA < 0.08, SRMR < 0.05). The model results indicate that there is a large direct effect of gender (0.425 standard deviations) on FMCE pre-score, but there is also an indirect effect of gender mediated by Test Anxiety and Sense of Belonging: female students have higher test anxiety (and lower belonging) and test anxiety in turn negatively affects FMCE pre-score (effect size = 0.057 standard deviations) while belonging positively affects FMCE pre-scores (effect size = 0.030). Furthermore, number

**Table 3. Multivariable linear regression predicting the effects of gender and high school physics coursework on exam scores.**

|  | Model 1 | Model 2 | Model 3 |
|---|---|---|---|
| Gender (F = 1) | -0.48*** (0.10) | -0.48*** (0.099) | -0.48*** (0.10) |
| No. Physics Courses |  | 0.31*** (0.057) |  |
| AP Physics 1 |  |  | 0.24* (0.11) |
| AP Physics C |  |  | 0.55*** (0.13) |
| R-squared | 0.054 | 0.12 | 0.093 |
| N | 366 | 366 | 366 |

*** p < 0.001

** p < 0.01

* p < 0.05.

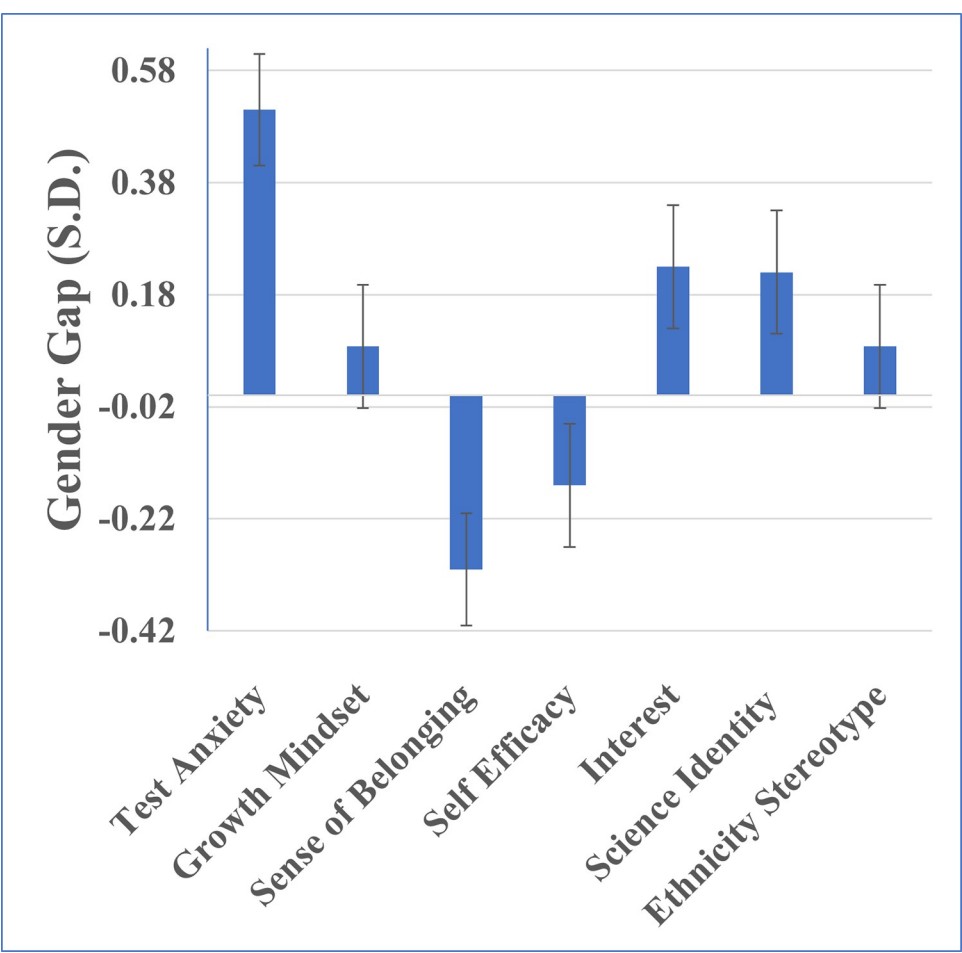

**Fig 2. Gender gaps on social psychological constructs.** Positive values indicate the female students score higher on that construct than male students. Error bars are the standard error of the coefficient. * $p < 0.05$, ** $p < 0.01$, *** $p < 0.001$.

of high school courses does not mediate the effect of Gender (effect size = 0.0070): while number of high school courses is correlated with FMCE scores, there is no gender difference in the number of high school courses. Contrarily, the positive benefits of female students' higher sense of science identity and interest in physics are small (indirect effects 0.015 standard deviations and 0.0083 standard deviations, respectively).

We triangulated the findings of the SEM model with multivariable regression (Table 4). The results are the same. There is a gender gap of 0.48 standard deviations which remains unchanged when you control for number of high school physics courses taken. Both sense of belonging and test anxiety are strongly correlated with FMCE scores, though the correlation with sense of belonging disappears when you account for Gender and HS coursework.

To look at the effect of all these variables and their interplay on exam performance, we extended the above mediation model by adding exam performance to the model. This model examines the potential effects of high school preparation, FMCE performance, Gender, and social-psychological factors on exam performance (Fig 4). All fit indices indicate that this mediation model is a good fit of the data covariance matrix.

Note that there is minimal change to all the coefficients present in Fig 2 that are also in Fig 3. We find that the FMCE pre-score is the sole mediator of the effects of Gender, social

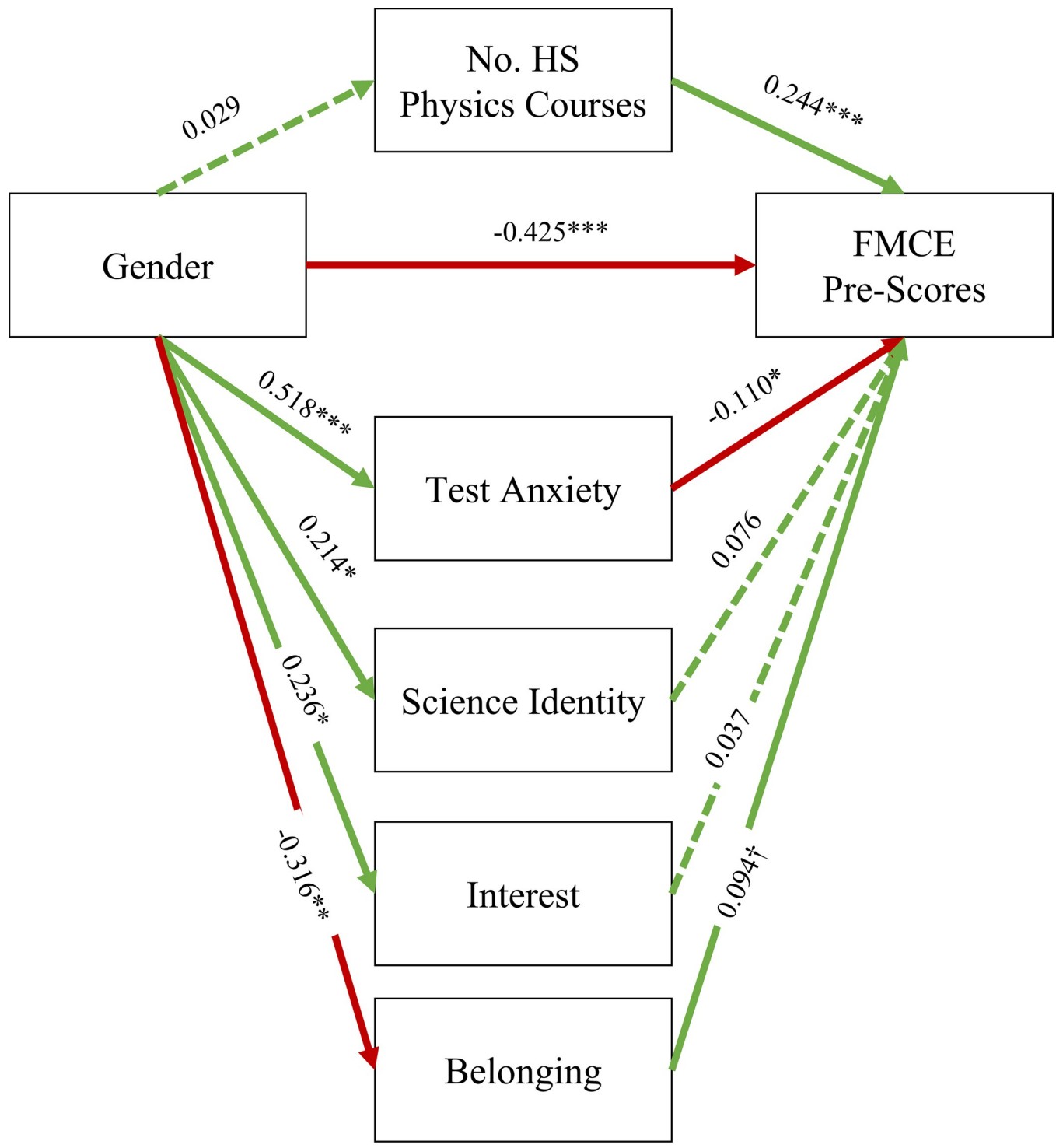

$$\chi^2(2) = 1.59, \text{p} = 0.452, \text{CFI} = 1.00, \text{TFI} = 1.00, \text{RMSEA} = 0.000, \text{SRMR} = 0.014$$

**Fig 3. SEM model of the effects of gender on FMCE pre-scores as mediated by number of high school physics courses, test anxiety, science identity, and interest in physics.** *** $p < 0.001$, ** $p < 0.01$, * $p < 0.05$, † $p < 0.10$. Acceptable ranges: CFI > 0.95, TLI > 0.90, RMSEA < 0.08, SRMR < 0.05 and p > 0.05 for whole model. Dashed links are not significant at the p = 0.10 level. Red links indicate negative effects, while green links indicate positive effects.

**Table 4. Multivariable linear regression predicting the effects of gender, high school physics coursework, and social psychological factors on FMCE scores.**

|  | Model 1 | Model 2 | Model 3 | Model 4 |
|---|---|---|---|---|
| Gender (F = 1) | -0.48*** (0.10) | -0.49*** (0.099) |  | -0.43*** (0.10) |
| High School Courses |  | 0.28*** (0.057) |  | 0.24*** (0.057) |
| Test Anxiety |  |  | -0.16** (0.052) | -0.11* (0.05) |
| Science Identity |  |  | 0.058 (0.059) | 0.076 (0.058) |
| Interest |  |  | 0.022 (0.052) | 0.037 (0.050) |
| Sense of Belonging |  |  | 0.15** (0.059) | 0.094 (0.057) |
| R-squared | 0.057 | 0.11 | 0.065 | 0.14 |
| N | 366 | 366 | 366 | 366 |

*** $p < 0.001$

** $p < 0.01$

* $p < 0.05$.

psychological factors, and high school preparation on exam performance (direct effect = 0.492 standard deviations, indirect effect = -0.009). In other words, test anxiety and belonging mediate the effect of gender on FMCE pre-scores, and FMCE pre-scores in turn predict exam performance. Therefore, a female and a male student with the same FMCE pre-score would obtain a similar exam score regardless of their physics courses and their social psychological orientation. Said more plainly, any effects of Gender, social-psychological factors, or high school physics courses on students' final exam performance is captured by the FMCE. We again triangulated these results with multivariable regression (Table 5). These results confirm that essentially all variables in exam performance are mediated by the FMCE pre-score.

Previous work shows that test-anxiety mediates the gender differences in **high-stakes timed** final exams. Here we see the similar effect of test anxiety on **zero-stakes timed** exam of FCME. Therefore, maybe it is plausible that lower performance of female students may be bound by the timed format of an assessment regardless of its weight. To test that, we look at the relationship between gender, FMCE pre-test, and performance on the first homework assignment, which is not a timed assessment like the final exam and FMCE (from 2018) using a regression analysis (Table 6). We find no gender gap on the first homework, while FMCE scores are correlated with homework 1 scores (indicating some dependence on high school preparation, $\beta$ = 0.14). The interaction between gender and FMCE pre-score is not significant and does not improve the fit of the model (ANOVA, p = 0.23), therefore, Model 2 is the best-fitting model in Table 4. Taken together with Fig 3, it seems likely the gender difference in final exam and pre-FMCE is due to their timed format partly as performance on timed assessments triggers test anxiety. Note that we did not perform a corresponding SEM for Table 6 because social psychological data was not available in 2018.

We note that HW1 covers much of the same content as the FMCE–basic 1D kinematics–thus it is notable that there is no gender gap on the assignment. Indeed, the assignment is still strongly predictive of final exam score (r = 0.26, p < 0.0001). This is not quite as strong as the correlation with FMCE score, likely because the FMCE covers a wider range of content and is thus more predictive of overall course performance. In prior work [3] we have seen similar patterns of the lack of gender gaps in measures that include homework scores.

## Discussion

There are several notable findings in this work. First, the size of the gender gap on the FMCE pre-test remains unchanged when controlling for prior physics coursework in high school.

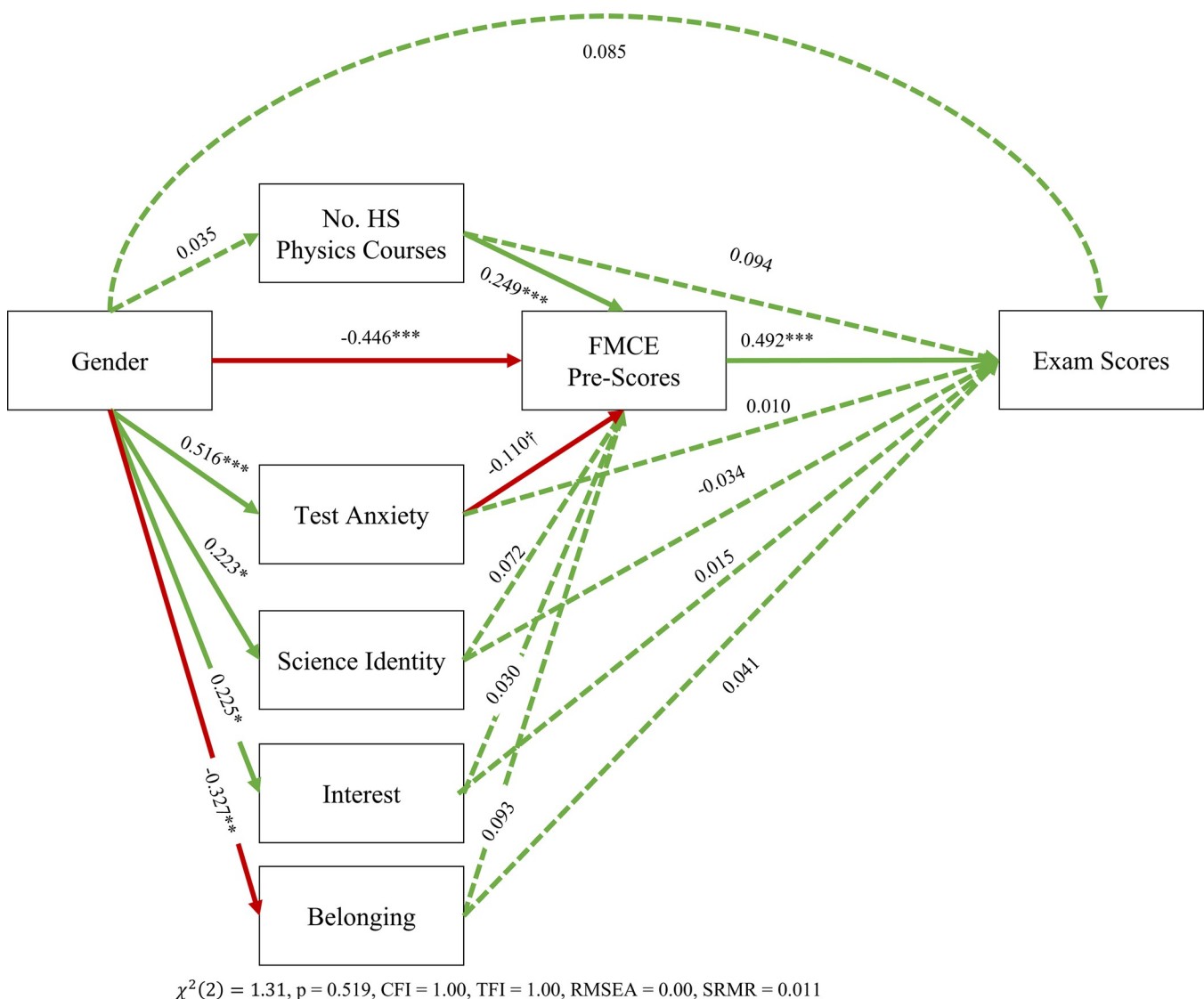

$\chi^2(2) = 1.31$, p = 0.519, CFI = 1.00, TFI = 1.00, RMSEA = 0.00, SRMR = 0.011

**Fig 4. Extension of the model in Fig 1.** We look at the effects of gender on final exam scores, as mediated by high school preparation, social-psychological factors, and performance on the FMCE. *** p < 0.001, ** p < 0.01, * p < 0.05, † p < 0.10. Acceptable ranges: CFI > 0.95, TLI > 0.90, RMSEA < 0.08, SRMR < 0.05 and p > 0.05 for whole model. Dashed links are not significant at the p = 0.10 level. Red links indicate negative effects, while green links indicate positive effects.

This is because, in this sample, there is no correlation between gender and a number or type of physics courses a student took in high school. Second, there is no gender gap on the first homework assignment, which should be similar to FMCE in its dependence on prior physics preparation, but different in format. Finally, we see that the gender gap in FMCE is partially explained by differences in female and male students' test anxiety and sense of belonging, even though the correlation between these factors and FMCE scores is small overall.

This leaves us with two potential explanations about the origin of the gender gap on the FMCE pre-test. First, it is possible that, though female students take physics in high school at similar rates as male students in this sample, they had negative experiences that impact their learning. This seems plausible given the reported influence of masculinity in physics [24] and the chilly or sometimes hostile climate that female students report in physics classrooms [25].

**Table 5. Multivariable linear regression predicting the effects of gender, social psychological variables, and FMCE pre-scores on exam grades.**

|  | Model 1 | Model 2 | Model 3 | Model 5 |
|---|---|---|---|---|
| Gender (F = 1) | -0.22* (0.10) | 0.093 (0.093) | 0.073 (0.096) | 0.085 (0.10) |
| FMCE Pre-score |  | 0.52***(0.046) | 0.49*** (0.049) | 0.49*** (0.050) |
| HS Courses |  |  | 0.095. (0.055) | 0.094. (0.056) |
| Test Anxiety |  |  |  | 0.010 (0.049) |
| Science Identity |  |  |  | -0.034 (0.055) |
| Interest |  |  |  | 0.015 (0.048) |
| Sense of Belonging |  |  |  | 0.041 (0.055) |
| R-squared | 0.0092 | 0.27 | 0.26 | 0.25 |
| N | 366 | 366 | 366 | 366 |

*** $p < 0.001$

** $p < 0.01$

* $p < 0.05$.

It is curious, however, that the female students who do persist to take physics in college have higher interest in physics and a higher sense of science identity. A plausible explanation is that these social-psychological motivators may have been necessary for the female students to overcome their negative experiences in high school physics courses and continue pursuing physics courses in college.

The second explanation is that the format of the FMCE as a "timed" closed book evaluation of physics competency induces stereotype threat in female students. As Spencer et al. [26] have shown, if an evaluation is presented to measure a competency of different groups in the topic, regardless of the stakes of the evaluation, the stereotyped group would underperform due to stereotype threat. They have also shown that anxiety is a mediator of the stereotype threat, and that stereotype threat is associated with lower sense of belonging. We observed the same pattern here. Even though the FMCE is zero-stakes, it is still framed as evaluative–assessing how much physics the students already know. This is likely to trigger stereotype threat. General test anxiety and sense of belonging partially mediate the gender gap in FMCE and explain 25% of the gap. Indeed, on the untimed first homework assignment, we see no gender gap even though homework grades also depend on prior physics preparation. Women suffer from the negative effect of stereotype threat despite higher sense of science identity and interest in Physics. In fact, Steele and colleagues have shown that stereotype threat is more consequential for the members of stereotyped groups who are more invested in the stereotype fields.

We see that the gender gaps in exam performance are entirely mediated by FMCE pre-scores, suggesting that any effects of test anxiety on final exam performance are captured by the timed nature of the FMCE. This would suggest that the stakes of the assessment do not

**Table 6. Regression model of homework 1 scores on FMCE pre-scores and gender.**

|  | Model 1 | Model 2 | Model 3 |
|---|---|---|---|
| Gender (F = 1) | 0.044 (0.098) | 0.11 (0.10) | 0.12 (0.10) |
| FMCE pre-score |  | 0.14** (0.052) | 0.091 (0.065) |
| Gender x FMCE |  |  | 0.12 (0.10) |
| R-squared | 0.00 | 0.014 | 0.015 |
| N | 518 | 518 | 518 |

All coefficients are standardized to be in units of standard deviations.

seem to matter as much as the fact that the assessment is timed. This factor appears to explain a portion of the gender gap, but a substantial amount of the gender gap is left unexplained by variables we collected. Because we find no gap in Homework 1 scores, we hypothesize that some other social-psychological measure might explain the remainder of this gap. There are many aspects of student thinking for which we do not have validated psychometric scales. For example, math anxiety is considered to be a separate construct from test anxiety and is reported to be higher in female students than male students. In the future, we will collect math anxiety measures to see if this explains part of the gap, as math is an important part of physics preparation and the FMCE contains many graphical interpretation questions.

There have been previous studies examining potential gender bias in physics conceptual assessments, including the FMCE, the instrument used here. In Henderson et al. [27], they conducted differential item functioning and found only one item which seemed to favor men over women. However, there are important methodological limitations to this finding. First, the DIF calculation assumes that the post-test concept inventory score is an accurate measure of students' ability. As our previous research has shown, this is less a measure of ability and more a measure of educational privilege, so the fundamental assumptions underlying this comparison are suspect. Indeed, the difference between men and women at post-test could be because of gender bias in the classroom itself, rather than actual understanding of physics knowledge. Even if this assumption held true, it would imply that intrinsic gender bias in this instrument is small compared to other factors like academic performance and social psychological effects. To the extent that our study overlaps with this investigation, it agrees with the finding that there are gender performance gaps.

A notable limitation to our study is that it was conducted at a highly selective university with substantial resources. We believe this largely explains why we see no difference in the number of previous physics courses men and women take–the university is highly selective and so all students are likely to take advanced science courses to stand out in the admissions process. However, we do not believe this invalidates our findings. Typically, at this university, the average score on most measures is higher than at other institutions, and the standard deviation is smaller. Thus, the amount of variance in performance outcomes that can be linked to academic preparation or test anxiety is likely lower in our sample than at other institutions. Regardless, the gender disparities in physics 1 exam and FMCE score is still present even in our selective sample. We have good reason to expect that test anxiety may play a larger role at other institutions where students have not been able to be as successful on admissions metrics like the SAT/ACT.

## Conclusion

This paper adds greater detail to the understanding of gender gaps in physics performance. While previous work had shown that gender gaps in exam performance and students' high school physics preparation, the source of those gender gaps are yet to be fully understood. Though nationally female students take physics at different rates, this did not explain the gender gap in physics preparation at this university. Despite the lack of gender difference in physics course taking behavior in high school, the work presented here shows that women in our sample are still at a disadvantage (obtain lower scores) in both FMCE and course final exam. This gender disparity is not due to lower physics preparation in high school as measured by number and type of high school physics courses taken by students, but more the result of social psychological phenomenon such as stereotype threat. There is no difference between men and women in number and type of high school physics courses. However, societal biases women suffer from negative stereotype in physics leads to their lower performance on FMCE and

exam as timed assessment of physics competency. This stereotype threat is partially mediated by general test-anxiety and to a small extent, sense of belonging.

We encourage other researchers to perform similar analyses with different student populations. We expect that some of the results (the lack of gender gap in physics course taking in high school, for example) are due to the highly selective nature of the population. Some of the other findings such as the gender gaps in sense of belonging and test anxiety are consistent with other reports, and so we believe are likely more general. It is unclear the extent to which test anxiety plays a role in assessment demographic gaps. We suspect this is *somewhat* sensitive to the nature of the ***assessment*** and population as shown in previous works [28], and thus may not be replicated for other assessment environments.

We conclude with some thoughts on how to avoid triggering this stereotype threat in college physics classrooms. The negative effect of stereotype threat can be diminished by changing the format of assessment. As we have shown here that there is no gender disparity in the first homework. The other works have also shown that women only under-perform on exams and, if anything, they overperform in other types of assessment [9,11]. This suggests that instructors should critically examine how they are assessing students' understanding of physics concepts. Timed, evaluative physics assessments which are framed to evaluate physics competency, even if low-stakes, are more likely to trigger stereotype threat in female students—in spite of (or because of) these students having higher science identity and interest in the topic. The effect of stereotype threat hampers their performance in these assessments, and even consequently can discourage them from pursuing physics-related STEM fields and have potentially long-lasting psychological effects on women as a marginalized group in STEM. This work adds to the ever-increasing body of education research works that call to transform our common assessment methods in STEM into more equitable formats that do not disadvantage marginalized and/or stereotyped groups.

## Author Contributions

**Conceptualization:** Eric Burkholder, Shima Salehi.

**Data curation:** Eric Burkholder.

**Formal analysis:** Eric Burkholder.

**Investigation:** Eric Burkholder, Shima Salehi.

**Methodology:** Eric Burkholder, Shima Salehi.

**Supervision:** Eric Burkholder.

**Writing – original draft:** Eric Burkholder, Shima Salehi.

**Writing – review & editing:** Eric Burkholder, Shima Salehi.

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
