## [Decision Letter · Decision Letter 0]

16 Mar 2022

PONE-D-22-04337Exploring the pre-instruction gender gap in physicsPLOS ONE

Dear Dr. Burkholder,

Thank you for submitting your manuscript to PLOS ONE. After careful consideration, we feel that it has merit but does not fully meet PLOS ONE’s publication criteria as it currently stands. Therefore, we invite you to submit a revised version of the manuscript that addresses the points raised during the review process.

The paper is competent and interesting. Some additional details (some stylistic) are needed-number of observations, correct specification of the empirical model. There are some more substantive clarifications sought by Reviewer 1-please attend to those so that we can take this forward.

We look forward to receiving your revised manuscript.

Kind regards,

Abhiroop Mukhopadhyay

Academic Editor

PLOS ONE

Journal Requirements:

Additional Editor Comments:

The paper is a competent and interesting contribution to the literature on gender gap in a STEM subject. The paper currently has some stylistic problems: missing mention of observations in tables, technically incorrect specification (lacking an error term in the equation). These are small things but important for scientific writing. In particular please pay special attention to comments of Reviewer 1.

Reviewers' comments:

Reviewer's Responses to Questions

**Comments to the Author**

1. Is the manuscript technically sound, and do the data support the conclusions?

Reviewer #1: Partly

Reviewer #2: Yes

2. Has the statistical analysis been performed appropriately and rigorously? 

Reviewer #1: No

Reviewer #2: Yes

3. Have the authors made all data underlying the findings in their manuscript fully available?

Reviewer #1: No

Reviewer #2: No

4. Is the manuscript presented in an intelligible fashion and written in standard English?

Reviewer #1: Yes

Reviewer #2: Yes

5. Review Comments to the Author

Reviewer #1: This is an interesting paper on the source of gender gaps in exam scores in an introductory physics course at university. The authors administer a test of physics-specific incoming preparedness (FMCE test) based on high school inputs to a sample of students in the very first class of an introductory physics course. They also survey the students to collect measures of social-psychological constructs such as test anxiety, sense of belonging, “science identity” and interest in physics. They model (using linear regression and structural equation modeling) the final exam scores of these students as a function of gender, FMCE test scores and social-psychological measures, and find that the entire gender gap in exam scores can be explained by a gender gap in the FMCE. They argue that both gender gaps in the FMCE and exam scores could arise out of timed-test anxiety, and find no evidence of gender gaps in non-timed tests.

I believe this paper makes some very interesting arguments but the empirical work could be made more transparent and clear, and the final claim on the impact of timed test anxiety could be further substantiated. Detailed comments and suggestions are enclosed in the attached report.

Reviewer #2: This is an interesting paper exploring the origins and drivers of gender gap in physics education using data from a specific educational institution. I have the following comments.

1. For international readers, it will be good to explain some of the terms specific to the US context, such as FMCE scores in the abstract, URM in introduction, SAT/ACT score in the introduction, R1 university in the data section, etc.

2. It seems that RQ1 and RQ2 discussed in the theoretical framework have already been explored in the literature and the current paper is just replicating the findings for the sample considered here. This aspect should be explicitly mentioned when the research questions are stated at the end of the theoretical framework section.

3. The presentation of the empirical analysis needs to improve.

a. The sample size is not given anywhere in the paper. I suggest that the estimation sample size is given for each regression (below R-squared).

b. There is no discussion on the external validity of these results. This could be addressed to some extent by discussing the nature of the sample -- what kind of population of students it represents.

c. The regression equations are not written accurately; if the error term is omitted, then it would mean that the explanatory variables included in the model perfectly predict the outcome, which is incorrect.

4. Since the main contribution of the paper relies on the measures described in Table 1, would it be possible to clarify these measures in the table itself, instead of mentioning the references only? That would make the paper self-sufficient and help the reader understand the measures directly from this paper.

6. PLOS authors have the option to publish the peer review history of their article (what does this mean?). If published, this will include your full peer review and any attached files.

Reviewer #1: No

Reviewer #2: **Yes: **Soham Sahoo

---

## [Decision Letter · Decision Letter 1]

27 Jun 2022

Exploring the pre-instruction gender gap in physics

PONE-D-22-04337R1

Dear Dr. Burkholder,

We’re pleased to inform you that your manuscript has been judged scientifically suitable for publication and will be formally accepted for publication once it meets all outstanding technical requirements.

Kind regards,

Abhiroop Mukhopadhyay

Academic Editor

PLOS ONE

Additional Editor Comments (optional):

You have addressed all queries of the reviewers. I have accepted the article but some very small typos remain-as pointed out by Reviewer 1. These should be addressed in the final version that is published.

Reviewers' comments:

Reviewer's Responses to Questions

**Comments to the Author**

1. If the authors have adequately addressed your comments raised in a previous round of review and you feel that this manuscript is now acceptable for publication, you may indicate that here to bypass the “Comments to the Author” section, enter your conflict of interest statement in the “Confidential to Editor” section, and submit your "Accept" recommendation.

Reviewer #1: All comments have been addressed

Reviewer #2: All comments have been addressed

2. Is the manuscript technically sound, and do the data support the conclusions?

Reviewer #1: Yes

Reviewer #2: Yes

3. Has the statistical analysis been performed appropriately and rigorously? 

Reviewer #1: Yes

Reviewer #2: Yes

4. Have the authors made all data underlying the findings in their manuscript fully available?

Reviewer #1: No

Reviewer #2: Yes

5. Is the manuscript presented in an intelligible fashion and written in standard English?

Reviewer #1: No

Reviewer #2: Yes

6. Review Comments to the Author

Reviewer #1: I thank the authors for responding to the comments and addressing most concerns. I believe this paper provides interesting insights on an important dimension of the gender gap in STEM. Minor comments and suggestions follow in the attached document.

Reviewer #2: (No Response)

7. PLOS authors have the option to publish the peer review history of their article (what does this mean?). If published, this will include your full peer review and any attached files.

Reviewer #1: No

Reviewer #2: No

---

## [Editor Report · Acceptance letter]

7 Jul 2022

PONE-D-22-04337R1 

Exploring the pre-instruction gender gap in physics 

Dear Dr. Burkholder:

I'm pleased to inform you that your manuscript has been deemed suitable for publication in PLOS ONE. Congratulations! Your manuscript is now with our production department. 

Kind regards, 

on behalf of

Dr. Abhiroop Mukhopadhyay 

Academic Editor

PLOS ONE